# Integrative epigenetic taxonomy of primary prostate cancer

Suzan Stelloo[1], Ekaterina Nevedomskaya[1,2], Yongsoo Kim[1,2], Karianne Schuurman[1], Eider Valle-Encinas[1], João Lobo [3,4,5], Oscar Krijgsman[6], Daniel Simon Peeper[6], Seiwon Laura Chang[7], Felix Yi-Chung Feng[7], Lodewyk Frederik Ary Wessels[2,8], Rui Henrique [3,4,5], Carmen Jerónimo [3,4,5], Andries Marinus Bergman[9,10] & Wilbert Zwart[1,11]

The Androgen Receptor (AR) is the key-driving transcription factor in prostate cancer, tightly controlled by epigenetic regulation. To date, most epigenetic profiling has been performed in cell lines or limited tissue samples. Here, to comprehensively study the epigenetic landscape, we perform RNA-seq with ChIP-seq for AR and histone modification marks (H3K27ac, H3K4me3, H3K27me3) in 100 primary prostate carcinomas. Integrative molecular subtyping of the five data streams revealed three major subtypes of which two were clearly TMPRSS2-ERG dictated. Importantly, we identify a third subtype with low chromatin binding and activity of AR, but with high activity of FGF and WNT signaling. While positive for neuroendocrine-hallmark genes, these tumors were copy number-neutral with low mutational burden, significantly depleted for genes characteristic of poor-outcome associated luminal B-subtype. We present a unique resource on transcriptional and epigenetic control in prostate cancer, revealing tight control of gene regulation differentially dictated by AR over three subtypes.

[1] Division of Oncogenomics, Oncode Institute, The Netherlands Cancer Institute, Amsterdam 1066 CX, The Netherlands. [2] Division of Molecular Carcinogenesis, Oncode Institute, The Netherlands Cancer Institute, Amsterdam 1066 CX, The Netherlands. [3] Cancer Biology and Epigenetics Group, Research Center of the Portuguese Oncology Institute-Porto, Porto 4200-072, Portugal. [4] Department of Pathology, Portuguese Oncology Institute of Porto, Porto 4200-072, Portugal. [5] Department of Pathology and Molecular Immunology, Institute of Biomedical Sciences Abel Salazar (ICBAS), University of Porto, Porto 4050-313, Portugal. [6] Division of Molecular Oncology & Immunology, Oncode Institute, The Netherlands Cancer Institute, Amsterdam 1066 CX, The Netherlands. [7] Department of Radiation Oncology, Helen Diller Family Comprehensive Cancer Center, University of California San Francisco (UCSF), San Francisco, CA 94115, USA. [8] Faculty of EEMCS, Delft University of Technology, Delft 2628 CD, The Netherlands. [9] Division of Medical Oncology, The Netherlands Cancer Institute, Amsterdam 1066 CX, The Netherlands. [10] Division of Oncogenomics, The Netherlands Cancer Institute, Amsterdam 1066 CX, The Netherlands. [11] Laboratory of Chemical Biology and Institute for Complex Molecular Systems, Department of Biomedical Engineering, Eindhoven University of Technology, PO Box 513, Eindhoven 5600 MB, The Netherlands. These authors contributed equally: Suzan Stelloo, Ekaterina Nevedomskaya, Yongsoo Kim.  Correspondence and requests for materials should be addressed to A.M.B. (email: a.bergman@nki.nl) or to W.Z. (email: w.zwart@nki.nl)

P rostate cancer is the second most common cancer in men worldwide, with over 1 million newly diagnosed cases each year[1]. Most men present with organ-confined prostate cancer which can potentially be cured through local therapy such as radical prostatectomy, radiation therapy, and/or brachytherapy[2]. However, approximately one-third of these patients will experience a rise in serum prostate-specific antigen (PSA), indicating cancer relapse which is referred to as biochemical recurrence[3]. Furthering our understanding of molecular alterations in primary prostate cancer might be helpful in determining why some patients develop a recurrence while others do not. To date, studies on primary prostate cancer have provided insight into the drivers of the disease, mostly focusing on Androgen Receptor (AR) function, mRNA expression, DNA copy number and mutations, as well as deviations in protein expression[4–10]. However, characterization of the epigenome in prostate cancer tissue is less well explored.

The most common genetic alteration in prostate adenocarcinomas is the fusion of members of the ETS transcription factor family (ERG, ETV1, and FLI1) with 5′ regions of androgen-responsive genes[8,11,12]. In particular, the TMPRSS2-ERG gene fusion is found in ~50% of the tumors, leading to over-expression of the oncogene ERG. Recently, it was shown that TMPRSS2-ERG fusion-positive tumors have a distinct H3K27ac chromatin landscape, facilitating ERG binding and recruitment of master transcription factors such as AR, FOXA1, and HOXB13 affecting the transcriptional program[13]. This reveals how genetic and epigenetic changes co-participate in regulating gene expression in TMPRSS2-ERG fusion-positive tumors. In addition to tumors with ETS fusions, the Cancer Genome Atlas (TCGA) network further classified primary prostate tumors in three additional molecular subtypes on the basis of mutations in SPOP, FOXA1, and IDH1. Both SPOP and FOXA1 are key regulators of the AR, and tumors carrying mutations in SPOP and FOXA1 possess enhanced AR transcriptional activity. However, the clinical significance of these mutations has not been thoroughly investigated. Others used gene sets associated with luminal and basal cell features to classify prostate cancers on outcome[14,15]. In the latter study[14], it was shown that luminal B classified tumors have the poorest clinical outcome but a better response to post-operative androgen deprivation therapy (ADT) than non-luminal B tumors.

To date, no prostate cancer studies have integrated genetic information together with epigenetic and gene expression data and stratified patients accordingly. Previously, we and others revealed distinct profiles of AR chromatin binding to classify patients on the outcome, comparing AR chromatin binding in benign or progressive prostate cancer with primary prostate cancer[16–18]. As AR—the main driver of prostate cancer —functions in conjunction with chromatin modifications to control transcription, we set out to comprehensively profile 100 primary prostate carcinomas by sequencing RNA transcripts in combination with ChIP-sequencing for AR and the active histone marks H3K27ac, H3K4me3 and repressive mark H3K27me3. Through multidimensional genomic data integration, we present three subtypes, of which two subtypes show a distinct chromatin landscape and transcriptional profiles determined by TMPRSS2-ERG fusion status.

## Results

**Study design and samples.** To investigate the potential existence of distinct epigenetic and transcriptomic features between primary prostate tumors, we compared RNA-seq and ChIP-seq profiles for 100 prostate cancer samples. As ChIP-seq variables, AR and histone modifications H3K27ac, H3K4me3, and H3K27me3 were studied. From a Porto cohort of 229 patients, we selected primary tumors of 49 patients who developed a biochemical recurrence within ~5 years after diagnosis and 50 samples without relapsed disease within ~10 years after diagnosis (Supplementary Figure 1). Samples were matched on age, initial PSA, T-stage, and Gleason score (Fig. 1a, Table 1). The median follow-up time for cases and controls was 153 and 150 months, respectively. In addition, two samples without follow-up data available were included in the study. For each data type, the following numbers of samples passed quality control: RNA-seq (n = 91), AR ChIP-seq (n = 88), H3K27ac ChIP-seq (n = 92), H3K4me3 ChIP-seq (n = 56), and H3K27me3 ChIP-seq (n = 76). For 42 samples, data was available for all genomic datastreams (Fig. 1b, Supplementary Figure 2). Information on ChIP-seq quality metrics is summarized in Supplementary Data 1 and Supplementary Figure 3. To validate the quality metrics of ChIP-seq, we utilized publicly available AR ChIP-seq (n = 13)[19] and H3K27ac ChIP-seq (n = 19)[13] on primary prostate cancers to compare normalized strand cross-correlation (NSC) and relative strand cross-correlation (RSC) values. Overall similar RSC and NSC values were observed among the datasets, with all NSC values from the ChIP-seq samples higher than input samples (Supplementary Figure 3c, d). The fraction of reads in peaks (FRiP) scores for H3K27ac in our study were lower as compared to those reported by Kron et al.[13] (Supplementary Data 2), which is possibly related to our lower sequencing depth[20]. As no input files were available for Pomerantz et al. samples[19], peak calling analyses could not be performed uniformly among both datasets, which affects FRiP score[20].

**Characterization of ChIP-seq data.** First, we visually inspected all AR, H3K27ac, H3K4me3, and H3K27me3 ChIP-seq profiles, as exemplified for four samples across expressed and repressed gene loci (Fig. 2a). Narrow peaks were observed for AR, H3K27ac, and H3K4me3 in contrast to H3K27me3 (Fig. 2b, c). The genome-wide ChIP-seq profiles for AR and the histone marks were highly distinct, dividing the samples into 4 clusters according to the factor ChIPped (Fig. 2d, e). Notably, the active histone marks (H3K27ac and H3K4me3) co-clustered (Fig. 2e, Supplementary Figure 4), which is in line with the described co-occurrence of these marks at promoters and transcribed regions[21]. AR binding is somewhat correlated with H3K27ac binding events, as expected due to its preferential binding at active enhancers (Fig. 2e, Supplementary Figure 4). Highest Pearson correlation coefficients were observed for H3K27ac samples, suggesting relative similarity of histone acetylation profiles between primary prostate cancer tissues (Fig. 2e, Supplementary Figure 4). However, more heterogeneous chromatin binding profiles were observed for H3K27me3 (Supplementary Figure 4). This is further supported by the steep decrease in the number of overlapping peaks as the number of samples increases for H3K27me3 ChIP-seq samples (Fig. 2f). To focus on the high-confidence peaks that were reproducibly identified in a large number of tumors, we considered those sites present in at least 25 out of 88 AR samples, 40 out of 92 H3K27ac samples, 25 out of 56 H3K4me3 samples, and 15 out of 76 H3K27me3 samples for further analysis. We found distinct genomic distributions: AR and H3K27me3 sites were mainly located at distal intergenic regions and introns (Fig. 2g). H3K4me3 signal was mostly enriched at promoters and introns, while H3K27ac was less enriched at promoters. This is in agreement with previously reported genomic distributions (AR[17–19], H3K27ac[13], H3K4me3 and H3K27me3[22]). For AR binding sites, DNA sequence motif analysis revealed, as expected, androgen response elements as well as forkhead motifs (Fig. 2h). We used publicly available

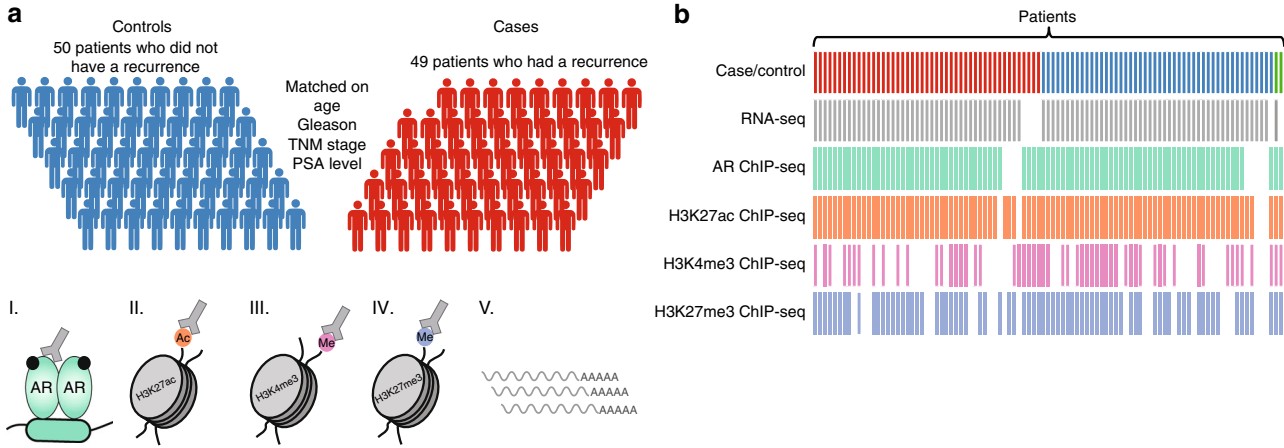

**Fig. 1** Overview of data. **a** Illustration of the matched case-control cohort. Prostate tumor samples from 49 patients with biochemical recurrence were matched to 50 patients without a recurrence. Samples were matched on clinical parameters age, TNM stage, PSA level, and Gleason score. Fresh frozen material of each patient was processed for RNA-seq, AR ChIP-seq, H3K27ac ChIP-seq, H3K4me3 ChIP-seq, and H3K27me3 ChIP-seq. The male silhouette was adapted from Wikipedia (https://upload.wikimedia.org/wikipedia/commons/archive/4/4e/20180727161732%21Aiga_toiletsq_men.svg). **b** Data availability for RNA-seq, AR ChIP-seq, H3K27ac ChIP-seq, H3K4me3 ChIP-seq, and H3K27me3 ChIP-seq for each sample. Columns represent individual patient samples. Cases and controls are depicted in red and blue, respectively. Two samples without follow-up are depicted in green (top row)

**Table 1 Characteristics of the Porto cohort (n = 101)**

|  | Cases n = 49 | Controls n = 50 | No follow-up n = 2 |
|---|---|---|---|
| Age at diagnosis | 64 (49–73) | 64 (47–72) | 67 (65–68) |
| PSA at diagnosis | 8.8 (3.3–23) | 8.2 (3.6–23) | 5.2 (4.5–5.8) |
| Gleason score |  |  |  |
| 6 | 6 | 14 | 1 |
| 7 | 28 | 22 | 1 |
| 8 | 3 | 5 | – |
| 9 | 12 | 9 | – |
| T stage |  |  |  |
| 2 | 15 | 15 | 1 |
| 3 | 33 | 35 | 1 |
| 4 | 1 | – | – |
| Lymph node status |  |  |  |
| N0 | 49 | 50 | 2 |
| N1 | – | – | – |
| Time to recurrence in months | 41 (14–74) | 120 (109–120) |  |
| Time to last follow-up in months | 153 (99–195) | 150 (70–191) | – |

AR ChIP-seq[19] and H3K27ac ChIP-seq[13] data to check for the enrichment of these factors at the consensus sites defined in this study. At these 8162 AR consensus sites and 31,085 H3K27ac consensus sites, robust and strong signal was found for all samples from the publicly available datasets (Supplementary Figure 5).

**Consensus clustering of RNA-seq and ChIP-seq data**. To discover prostate cancer subgroups with distinct epigenetic and transcriptomic profiles, we selected the top most-variable genes based on RNA-seq and the most-variable regions among the high confidence peaks of AR, H3K27ac, H3K4me3, and H3K27me3 ChIP-seq data across the samples. Consensus hierarchical clustering[23] was performed on each dataset and revealed the presence of five stable patient clusters in RNA-seq data (Fig. 3a). Based on ChIP-seq, using the normalized read counts in peaks, three patient clusters were identified using AR and H3K27me3 profiles and two patient clusters were found using H3K27ac and H3K4me3 data (Fig. 3a). The heatmaps of consensus matrices

that show the stability of clustering for the number of chosen k-clusters for each datatype are shown in Supplementary Figure 6. The resultant clusters do not appear to substantially separate cases and controls in both transcriptomic and ChIP-seq data (Fig. 3a, b), which was confirmed in principal component analysis (Supplementary Figure 7). RNA-seq clusters 1 and 2 (dark blue and light green clusters) were relatively enriched for high Gleason score (p-value = 0.005), while no differences in Gleason score were observed for all ChIP-seq based clusters. Samples were further classified into luminal and basal-like subtype using the PAM50 classifier, which was recently applied successfully in prostate cancer[14] (Supplementary Figure 8). Notably, RNA-seq clusters 1 and 2 (dark blue and light green clusters) were mainly comprised of luminal B tumors, while clusters 4 and 5 (light blue and dark green clusters) contained the majority of samples classified as luminal A (p-value = 6.83e−06). However, the PAM50 subtypes distributed almost evenly between all clusters in the ChIP-seq datasets. Next, we analyzed AR pathway activation within the tumors using a previously reported AR activity signature[24], which was also implemented in the TCGA report on primary prostate tumors[8]. The AR activity score was highly variable among tumors (range: −23.5 to 19.0). Interestingly, AR activity score is associated with RNA-seq, H3K4me3- as well as H3K27me3-derived clustering (p-value = 7.64e−05, 0.0006, 0.005, respectively) (Fig. 3b). Given that ERG fusion status represents one of the largest molecular subtypes in the TCGA prostate cancer cohort[8], we analyzed ERG mRNA expression in our patient population. We observed a bimodal distribution of ERG expression, with 45 samples showing low and 46 samples high ERG expression, suggesting ERG translocations in high ERG expressing samples (Supplementary Figure 9a). To confirm expression of ERG fusion transcripts in high ERG expressing tumors, we examined the presence of fusion junction spanning reads in our RNA-seq data, as well as the 5 prime–3 prime ERG transcript ratio. Furthermore, a fraction of the tumors was previously assessed for ERG rearrangements[12]. Indeed, we only observed junction reads and high 5 prime–3 prime transcript ratio in high ERG expressing tumors, which were strongly concordant with ERG rearrangement status (Supplementary Figure 9b, c). This prevalence of ERG fusions in ~50% of the tumors is consistent with other studies[8]. Other than ERG fusions, we observed nine tumors with ETV1 rearrangements (10%) and 1

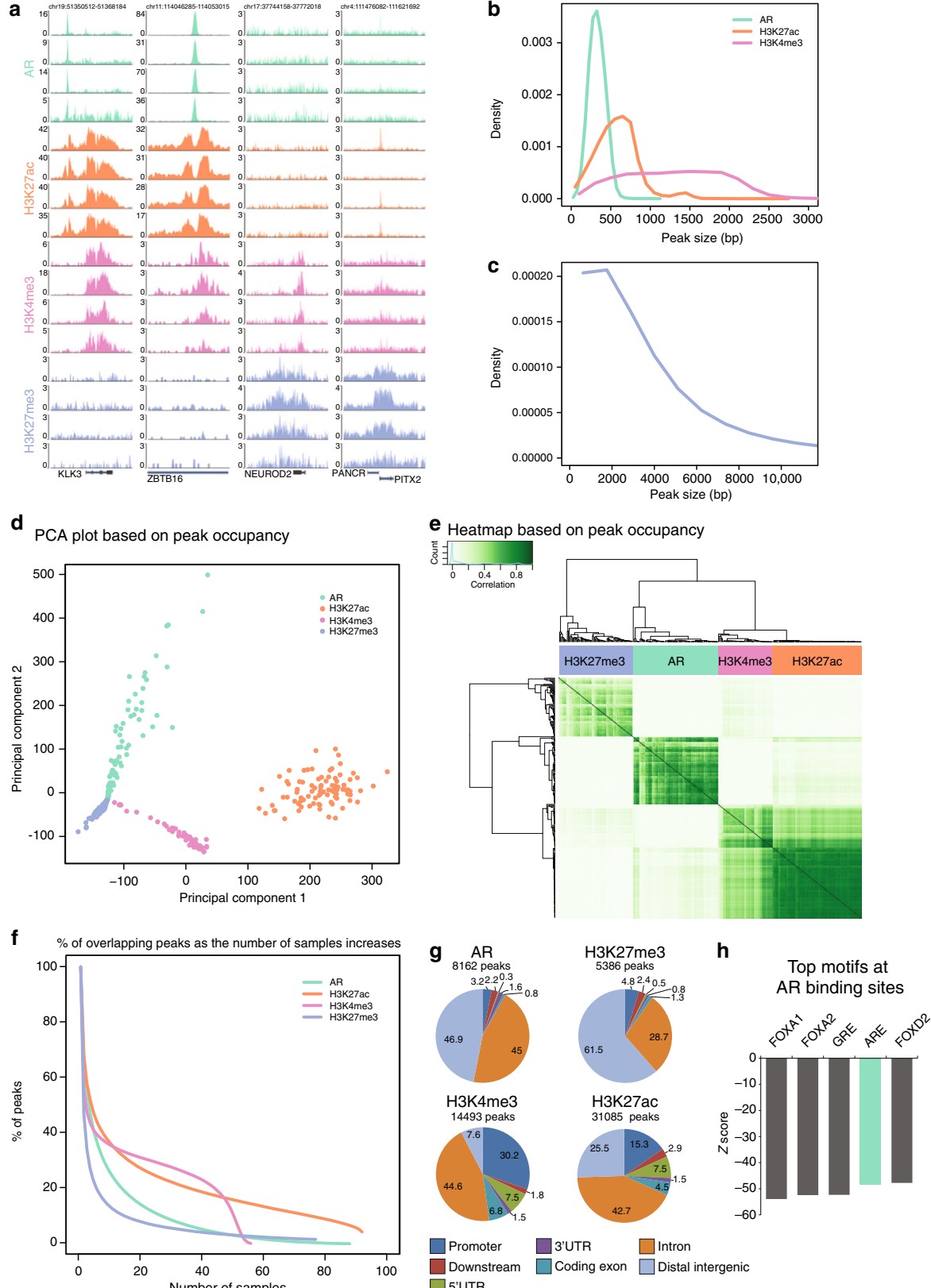

tumor with an *ETV5* fusion transcript (1%). Transcriptome-wide analysis between low and high ERG expressing tumors revealed differential expression of previously identified ERG target genes (Supplementary Figure 9d), and two gene sets related to *TMPRSS2-ERG* fusion from MSigDB show significant enrichment

(Supplementary Figure 9e). Remarkably, the five mRNA clusters nearly perfectly matched ERG transcript classification, capturing three clusters with low ERG expressing tumors (clusters 2, 3, 4; light green, salmon, light blue) and two clusters with high ERG expressing samples (cluster 1; dark blue and cluster 5; dark green)

**Fig. 2** Characterization of ChIP-seq data. **a** Snapshots for AR (green), H3K27ac (orange), H3K4me3 (pink), and H3K27me3 ChIP-seq are shown at four example loci in four patients. Genomic coordinates are indicated above. **b** Peak width distribution of ChIP-seq peaks for AR (green), H3K27ac (orange), and H3K4me3 (pink) peaks. **c** Distribution of peak width for H3K27me3 ChIP-seq peaks. **d** Scores plot of principal component analysis based on occupancy (called peaks) of AR (green), H3K27ac (orange), H3K4me3 (pink), and H3K27me3 (purple) ChIP-seq samples. **e** Correlation heatmap based on peak occupancy. The clustering of the samples represents correlations between individual ChIP-seq samples on the basis of all called peaks. The column color bar indicates the ChIPped factor. Pearson correlation is plotted in white-green color scale. **f** Plot depicts the number of peaks overlapping in tumors for each factor ChIPped. Consensus peakset were chosen by using a cutoff of peaks present in at least 25, 40, 25, or 15 samples for AR, H3K27ac, H3K4me3, and H3K27me3, respectively. The number of consensus peaks is indicated for each factor. **g** Genomic distribution of consensus peaks from AR, H3K27me3, H3K27ac, and H3K4me3 across genomic features. **h** Bar chart shows the $Z$-score of the top 5 sequence motifs found at consensus AR peaks

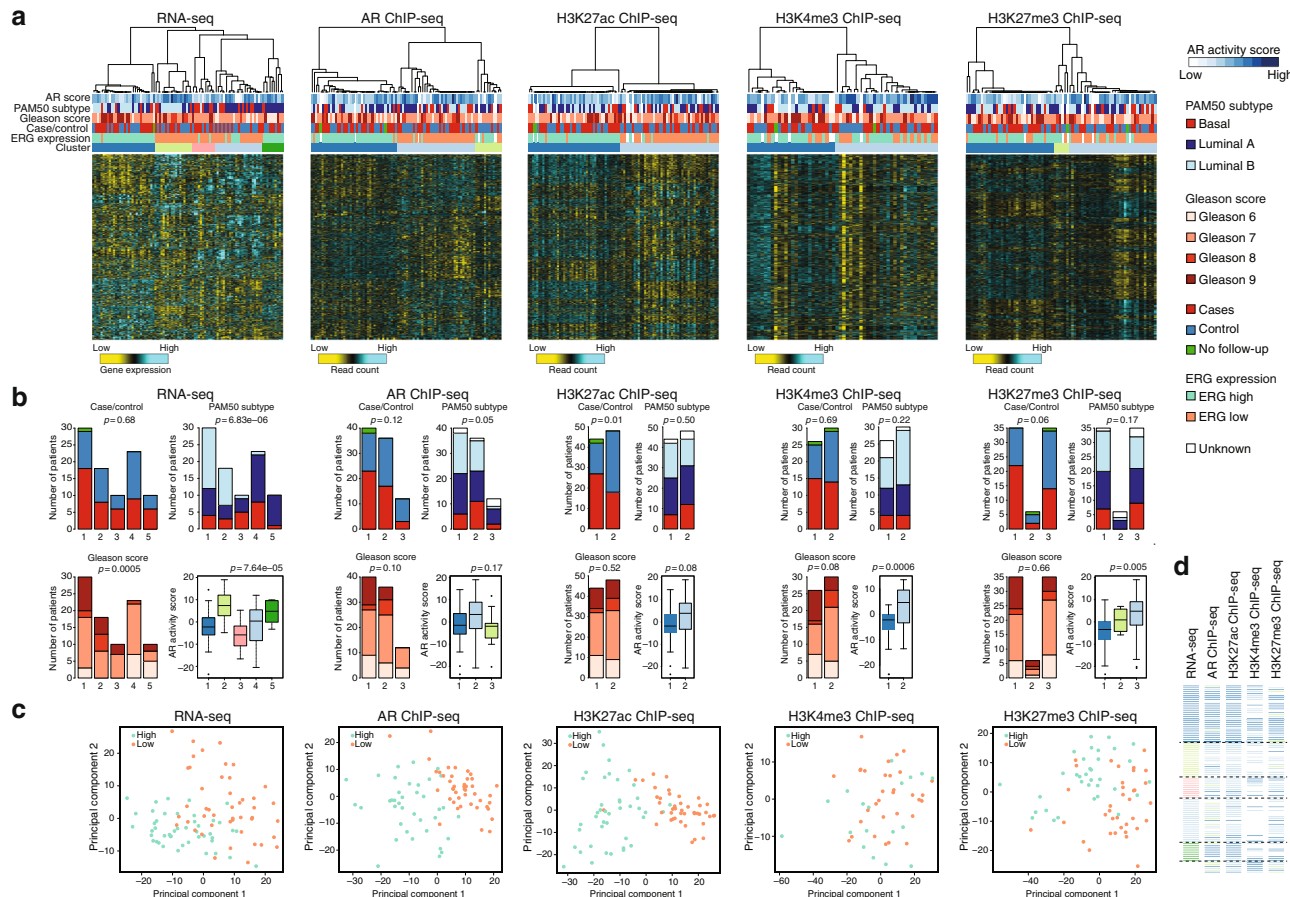

**Fig. 3** Consensus clustering of prostate cancer samples by each dataset. **a** Heatmaps of RNA expression, AR binding, H3K27ac, H3K4me3, and H3K27me3 ChIP-seq signal. Each sample is annotated for AR activity score, PAM50 subtype, Gleason score, case/control status, ERG expression, and consensus cluster assignment. Samples are ordered according to consensus clustering in Supplementary Figure 6. As shown in the color scale for RNA-seq, yellow indicates relatively low expression and blue relatively high expression ($z$-scores), whereas for ChIP-seq, yellow indicates relatively low peak intensity and blue relatively high peak intensity ($z$-scores). **b** Plots showing the distribution of case/control status, Gleason score, PAM50 subtypes, and AR activity score across the clusters identified in RNA-seq, AR ChIP-seq, H3K27ac ChIP-seq, H3K4me3 ChIP-seq, and H3K27me3 ChIP-seq datasets. Boxplots represent median AR activity scores with interquartile ranges. **c** PCA scores plot for RNA expression, AR binding, H3K27ac, H3K4me3, and H3K27me3 ChIP-seq signal, based on the top 1000 most-varying genes/regions across the samples. Samples are colored according to ERG high or low expression. **d** Comparison of consensus cluster assignment of the samples (rows) for each datatype (columns)

(Fig. 3a). Also, for all ChIP-seq based clustering, except for H3K4me3, strong separation on ERG expression was found. We confirmed the strong association with ERG expression using principal component analyses on each molecular platform, which revealed separation between low and high ERG expressing tumors, with exception of H3K4me3 ChIP-seq (Fig. 3c). We also found separation of ERG fusion positive and negative samples using previously published H3K27ac ChIP-seq data looking at the sites defined in this study (Supplementary Figure 10a). The sites defined in the previous study, also classified the Porto samples on ERG expression (Supplementary Figure 10b). In conclusion, clustering based on both transcriptomic and epigenetic profiles identifies distinct groups that largely overlap with ERG classification.

Individual ChIP-seq data clustering shows similarities to transcriptomic clustering (Fig. 3d), which likely reflects the interplay between epigenetic modifications and gene expression. Indeed, the presence of AR and the chromatin marks H3K27ac and H3K4me3 are associated with active gene transcription, while H3K27me3-marked genes have low expression level (Supplementary Figure 11).

Noteworthy, low concordance between H3K4me3-based clustering and any other clustering (RNA-seq, ChIP-seq, ERG expression) was observed, which may be in part affected by the smaller sample size for H3K4me3 ChIP-seq. To capture both concordant and unique features across the individual data types, we next performed integrative molecular clustering.

**Integrative epigenomic analysis identifies three subtypes.** To perform integrative molecular characterization, we applied clustering combining available data from RNA-seq and ChIP-seq on 97 prostate cancers using multiple incomplete-view non-negative matrix factorization (MIV-NMF). This computational approach allows for the use of incomplete data, so that all samples (including those for which one or more datastream are missing) could be included in the analyses. To find the number of clusters that provide a stable solution, we applied MIV-NMF multiple times with random initialization using varying number of clusters from $k = 2$ up to $k = 5$. Consensus scores for different $k$'s suggested robust assignment of the samples into two or three clusters (Fig. 4a and Supplementary Figure 12). For $k = 2$, the samples are basically divided into an ERG low and an ERG high group, which is in line with the individual clustering. Intriguingly, within the three cluster solution, $k = 3$, an additional group was revealed that was not dictated by ERG expression. Repeated clustering for

100 times with 80% random subsampling identified similar cluster solution, emphasizing the stability of the three clusters (Supplementary Figure 13). Motivated by the identification of the third subcluster, we focused on the three cluster solution; Cl1 ($n = 40$), Cl2 ($n = 28$), Cl3 ($n = 29$) (Fig. 4a).

We assessed the contribution of each data type in integrative clustering analysis by comparing consensus factor matrix to individual data factor matrix derived from MIV-NMF analysis. RNA-seq data achieved the highest Pearson's correlation coefficient of 0.63, while H3K4me3 contributed least (Pearson's correlation coefficient = 0.11) (Fig. 4b). The integrative clustering-produced classifications were highly concordant with the individual clustering classifications (Fig. 4c). Importantly, no individual molecular data type could recover all three cluster of integrative clustering analysis, providing compelling evidence that the five data types contain complementary information.

**Characterization of the three subtypes.** To explore reproducibility of the three prostate cancer subtypes, we downloaded gene expression data for 497 primary prostate cancers from the TCGA database. The top 100 most differentially expressed genes in each of the three clusters in Porto data (in total 285 genes, Supplementary Data 3) were used to perform unsupervised hierarchical clustering on mRNA data from the TCGA cohort. As in our

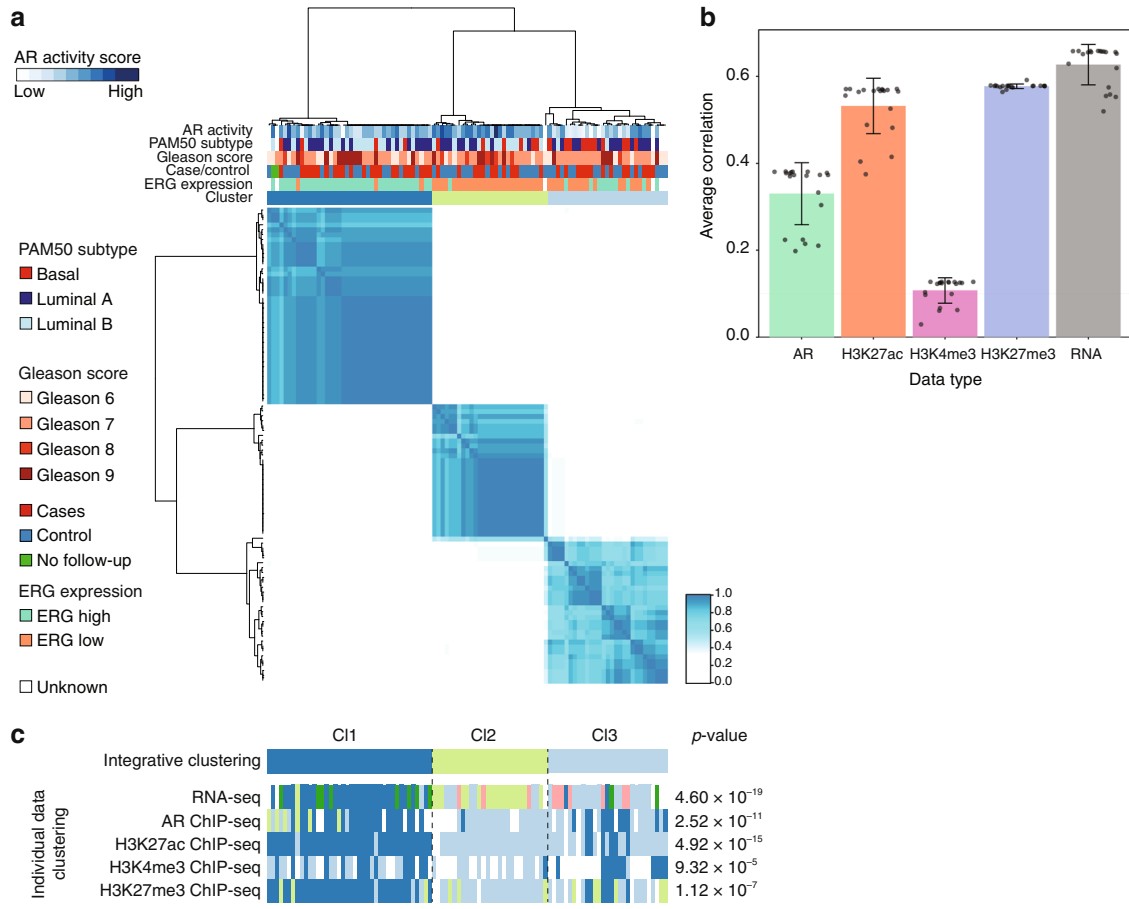

**Fig. 4** Integrative clustering of prostate cancer samples. **a** Heatmap displays the consensus matrix of integrative analysis (MIV-NMF) on the basis of RNA-seq, AR, H3K27ac, H3K4me3, and H3K27me3 ChIP-seq for $k = 3$ (three clusters). Rows and columns are samples, and the more frequently samples occur in the same cluster, the darker the color blue. **b** Bar plot showing the Pearson's correlation coefficient capturing the contribution of each data type in integrative clustering. Error bars indicate the standard deviation. **c** The sample classification into three clusters according to integrative clustering analysis is shown in the top row. The bottom panels show the resultant clustering according to consensus clustering analysis based on each individual data type sorted by integrative cluster assignment. Chi-square tests were used to test for association between the integrative clusters and clusters obtained in each individual data type

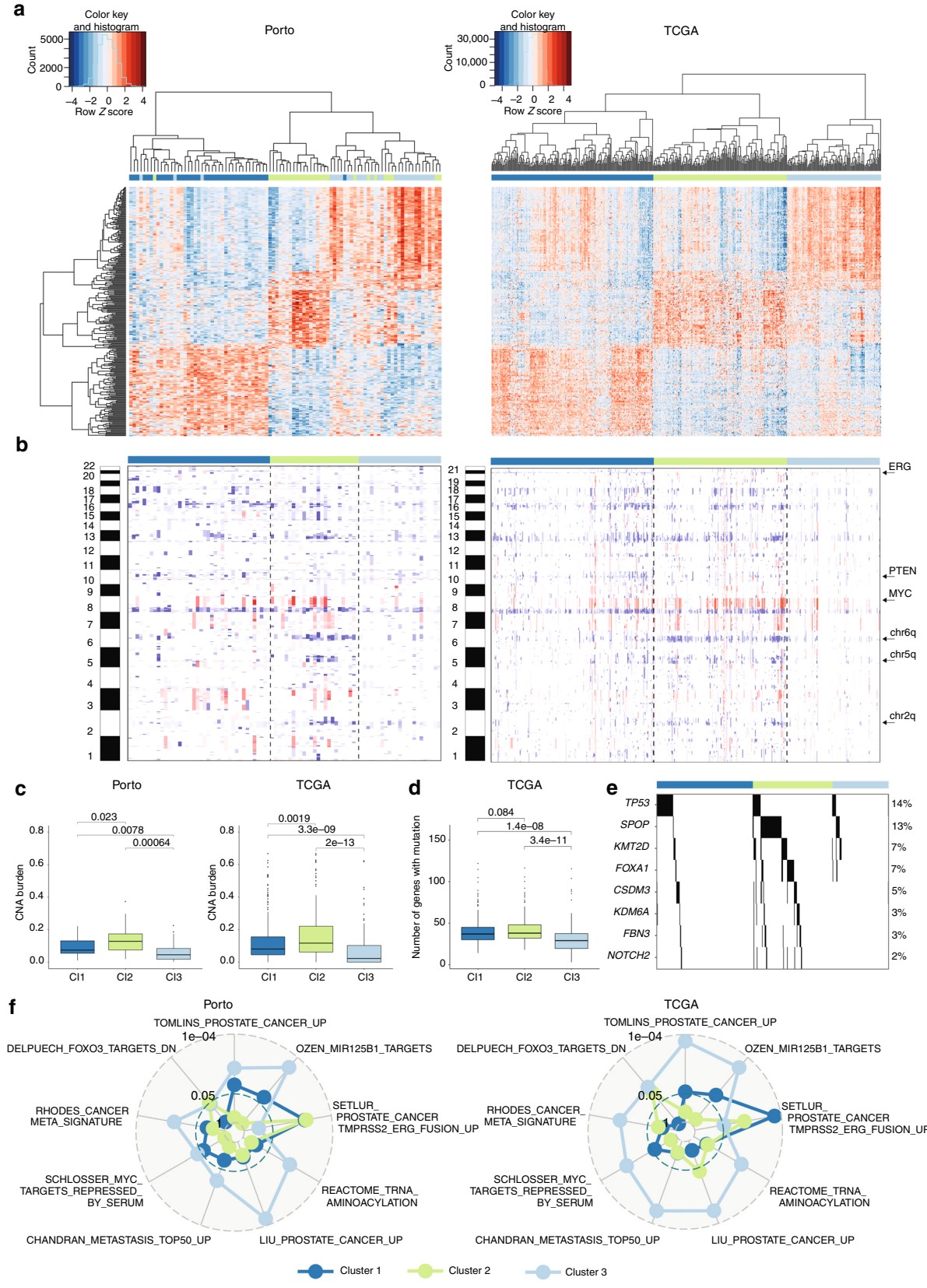

dataset, the TCGA dataset is partitioned into three clusters with comparable expression profiles as the three clusters in the Porto cohort (Fig. 5a). The three identified clusters in both the Porto as well as the TCGA cohort will be termed hereafter as cluster 1 (Cl1), cluster 2 (Cl2), and cluster 3 (Cl3).

Tables 2 and 3 show associations between the three clusters identified in the Porto cohort and the TCGA cohort, respectively, with clinical data, ERG expression, PAM50 classification, and AR activity score. Most of the clinical variables, including age, initial PSA, T-Stage, Gleason score, and biochemical recurrence, did not

**Fig. 5** Validation and characterization of three subtypes. **a** On the left: unsupervised hierarchical clustering of 285 differentially expressed genes between the three integrative clusters. The sample classification according to integrative cluster analysis is indicated below the branching. On the right: unsupervised hierarchical clustering on the TCGA cohort using the genes differentially expressed across the three clusters of the Porto cohort. The genes (rows) in the heatmap are ordered the same as for the Porto cohort. Color scale: red indicates high expression an blue low expression (z-score). **b** Heatmap of copy number alterations (CNAs) of 88 Porto samples (left) and TCGA samples (right). Samples from Porto and TCGA cohort are ordered the same as in Figs. 4a and 5a, respectively. Red and blue represent copy number gains and losses, respectively. **c** CNA burden as fraction of the genome that is copy number altered in the Porto cohort (left) and TCGA cohort (right). Boxplot: median values with interquartile range. p-Values were calculated using the Wilcoxon test. **d** Boxplot (median values with interquartile range) showing the number of genes with a mutation for the three clusters in the TCGA cohort. p-Values were calculated using the Wilcoxon test. **e** Association of mutation with the three clusters identified in the TCGA cohort. Only genes with significant differential enrichment among the clusters are shown (FDR < 0.2). Percentages on the left show the mutation frequency. **f** p-Values for top gene sets enriched (FDR < 0.2) from the MSigDB collection of "curated" gene sets in any of the three clusters, represented in a radar plot for Porto (left) and TCGA cohort (right). p-Value for each cluster is indicated with a line with the corresponding color

### Table 2 Patient characteristics by integrative clusters—Porto cohort

| | Integrative clustering—Porto cohort | | | |
| | CI1 $n = 40$ | CI2 $n = 28$ | CI3 $n = 29$ | p-Value |
|---|---|---|---|---|
| Age at diagnosis | 63.2 | 63.3 | 63.4 | 0.984 |
| PSA at diagnosis | 8.94 | 9.54 | 9.94 | 0.648 |
| T stage | | | | 0.932 |
| 2 | 8 | 10 | 10 | |
| 3 | 31 | 18 | 19 | |
| 4 | 1 | – | – | |
| Gleason score | | | | 0.357 |
| 6 | 10 | 4 | 6 | |
| 7 | 18 | 14 | 17 | |
| 8 | 2 | 5 | 1 | |
| 9 | 10 | 5 | 5 | |
| Group | | | | 0.224 |
| Case | 23 | 11 | 14 | |
| Control | 15 | 17 | 15 | |
| Exclude | 2 | – | – | |
| ERG status | | | | $6.642 \times 10^{-12}$ |
| Low | 3 | 26 | 16 | |
| High | 35 | 1 | 10 | |
| NA | 2 | 1 | 3 | |
| PAM50 subtype | | | | 0.01343 |
| Basal | 4 | 6 | 11 | |
| LumA | 17 | 9 | 13 | |
| LumB | 17 | 12 | 2 | |
| NA | 2 | 1 | 3 | |
| AR activity | −0.97 | 4.97 | −4.94 | 6.825e−6 |
| Clinical recurrence | | | | 0.2379 |
| No | 28 | 24 | 24 | |
| Yes | 12 | 4 | 5 | |
| Death | | | | 0.07688 |
| No | 30 | 18 | 26 | |
| Yes | 10 | 10 | 3 | |
| Cause of death PCa | | | | 0.3995 |
| No | 35 | 27 | 27 | |
| Yes | 5 | 1 | 2 | |

### Table 3 Patient characteristics by three clusters—TCGA cohort

| | TCGA clustering | | | |
| | CI1 $n = 206$ | CI2 $n = 171$ | CI3 $n = 120$ | p-Value |
|---|---|---|---|---|
| Age at diagnosis | 61 | 63 | 61 | 0.025 |
| PSA at diagnosis | 7.2 | 9.6 | 5.7 | 0.7587 |
| T stage | | | | 0.530 |
| 2 | 71 | 67 | 49 | |
| 3 | 130 | 98 | 65 | |
| 4 | 3 | 5 | 2 | |
| NA | 2 | 1 | 4 | |
| Gleason score | | | | 0.834 |
| 6 | 17 | 14 | 14 | |
| 7 | 101 | 82 | 64 | |
| 8 | 26 | 23 | 15 | |
| 9 | 61 | 50 | 26 | |
| 10 | 1 | 2 | 1 | |
| Biochemical recurrence | | | | 0.312 |
| No | 159 | 120 | 92 | |
| Yes | 25 | 20 | 13 | |
| NA | 22 | 31 | 15 | |
| ERG status | | | | <2.2e−16 |
| Low | 26 | 170 | 100 | |
| High | 180 | 1 | 20 | |
| PAM50 subtype | | | | 7.2e−15 |
| Basal | 36 | 25 | 34 | |
| LumA | 82 | 57 | 80 | |
| LumB | 88 | 89 | 6 | |
| AR activity | −0.72 | 5.36 | −1.31 | <2.2e−16 |
| RFS event | | | | 0.7029 |
| 0 | 158 | 126 | 100 | |
| 1 | 24 | 17 | 11 | |
| NA | 24 | 28 | 9 | |

*RFS* recurrence-free survival

show significant differences across the three clusters in both cohorts. In the Porto cohort, out of the 97 patients, 21 patients developed a clinical recurrence, 23 patients died of which 8 died of prostate cancer. Between the three subgroups, no statistically significant differences could be observed on metastatic-free survival and overall survival, which may be related to the low number of events. In the TCGA cluster 2 however, the age at diagnosis is slightly higher than cluster 1 and cluster 3 (p-value = 0.025). As expected from individual data cluster analysis, the three clusters were strongly associated with ERG expression status

in both cohorts (Porto cohort; p-value = 6.642e−12, TCGA cohort; p-value < 2.2e−16). Tumors in cluster 1 are characterized by high ERG expression, while tumors in cluster 2 express low ERG levels. Interestingly, ERG levels were not the sole-driving factor, as cluster 3 contains both tumors with low and high ERG expression level. Assessment of PAM50 subtype classification for the three newly-identified clusters in both cohorts, revealed that luminal B tumors were less frequently classified into cluster 3, while basal-like tumors were less often grouped into cluster 1. Also, AR activity score was associated with the three clusters (Porto cohort; p-value = 6.825e−6, TCGA cohort; p-value < 2.2e−16), revealing high AR activity score in cluster 2 and a low score in cluster 3. Though AR mRNA expression levels were slightly

lower in cluster 3 (Supplementary Figure 14a), but no difference in AR protein expression was observed among the clusters (Supplementary Figure 14b). These data imply that not AR expression levels, but its activation status was diminished selectively in cluster 3. These results were further confirmed by AR ChIP-seq data, as the number of AR peaks was lower in cluster 3 (Supplementary Figure 14c), while the number of peaks for H3K27ac did not differ between the clusters (Supplementary Figure 14d). As low AR activity is observed in prostate cancers with a neuroendocrine phenotype[25], we analyzed the expression of a 70-gene neuroendocrine prostate cancer (NEPC) classifier[26]. Intriguingly, tumors in clusters 1 and 2 scored low on NEPC-likeness, while cluster 3 tumors showed a more NEPC-like expression profile (Supplementary Figure 14e). To detect neuroendocrine differentiation on the protein level, we performed immunohistochemistry for the detection of two neuroendocrine markers typically used in clinical practice, chromogranin and synaptophysin. As both genes (CHGA and SYP) are not represented in the neuroendocrine gene expression signature, these findings would be an independent validation of NEPC-status in these tumors. However, most cases have less than 1% of cells expressing chromogranin and/or synaptophysin (Supplementary Figure 15a). Only 10 samples stained positive (>1%) for chromogranin and 22 samples for synaptophysin. Importantly, both CHGA and SYP were not enriched in cluster 3 at either the mRNA level and protein level (Supplementary Figure 15b, c). Thus, while cluster 3 was enriched for tumors that possess a number of neuroendocrine-like features, including lack of AR functionality (even though the receptor is expressed) and enrichment for a neuroendocrine gene signature, classical pathological neuroendocrine markers were negative in these tumors.

We next sought to relate the three clusters to copy number alterations. In the Porto cohort, we determined copy number alterations by using CopywriteR[27] that utilizes the off-target reads in AR ChIP-seq data. We identified several well-known copy number alterations, including loss of chromosome arm 8p, 10q and chromosome 8q gain (Fig. 5b). Notably, in both Porto and TCGA cohorts, the tumors in cluster 3 tend to have minimal copy number alterations, in contrast to the tumors in cluster 2 that showed most copy number alterations (Fig. 5b, c). The samples in TCGA cluster 3 showed mildly lower tumor percentage which reached statistical significance (Supplementary Figure 14f), but this was not the case for the Porto cohort, excluding tumor cell percentage as a sole driver in these findings. Cluster 2 is mainly distinguished by the alterations at chromosomes 2q, 5q, 6q, and 8q (Fig. 5b). As cluster 1 comprises tumors with a TMPRSS2-ERG fusion, this cluster exhibited an interstitial deletion between the two genes on chromosome 21. In addition, these samples can be further characterized by loss of chromosome 10q (PTEN) and 17p (TP53), previously shown to be associated with TMPRSS2-ERG fusions[5].

The samples in the TCGA cohort have been previously characterized for mutations in coding regions. A low mutational burden was observed for tumors classified in cluster 3 as compared to tumors in clusters 1 and 2 (Fig. 5d). Mutations with significant enrichment in one of the clusters are shown in Fig. 5e. Mutations in TP53 are enriched in cluster 1, while mutations in SPOP, KMT2D, FOXA1, CSDM3, KDM6A, FBN3, and NOTCH2 are enriched in cluster 2. Among the mutated genes in cluster 2, FOXA1 and SPOP mutations have been previously shown to be mutually exclusive with ERG fusions possessing high AR activity[8,28,29]. Importantly, none of the mutations are enriched in cluster 3.

Finally, to identify gene sets and regulatory networks associated with the molecular subtypes, we performed differential gene expression analysis between the three clusters within both the Porto and TCGA cohorts. Even though most of the gene sets reached significance only in the TCGA cohort, most likely due to smaller sample size of the Porto cohort, the normalized enrichment score (NES) between the Porto and TCGA cohort correlated well. In cluster 1, comprising ERG fusion-positive tumors, only one gene set related to ERG fusion passed FDR threshold < 0.2 (Fig. 5f, Supplementary Figure 16a). In cluster 3, we observed gene sets reflecting benign-like state among the most significantly deregulated gene sets. The top gene sets are shown in Fig. 5f (see Supplementary Data 4 for all significantly deregulated gene sets). To narrow-down the key characteristics for cluster 3, we identified differentially expressed regulatory networks by integrating transcriptomics data with protein–protein interaction datasets using HotNet2[30]. HotNet2 identified eight large networks including those indicating activated fibroblast growth factor (FGF) signaling, WNT signaling, and nerve growth factor (NGF) signaling (Supplementary Figure 16b and Supplementary Data 5). As cluster 3 was hallmarked by inactive AR signaling, strong activation of the above-mentioned pathways may indicate dependence on alternative drivers for this newly identified patient subpopulation. Recently, single cell RNA-seq analyses in CTCs indicate WNT signaling activation in antiandrogen resistant tumors, supporting functional compensation by WNT in absence AR action, albeit in the progressive disease setting[31]. Also, FGF signaling pathway has been reported as an alternative driver to bypass AR dependence[32].

Cumulatively, these data indicate that by integrative analyses, we revealed a subclass of prostate tumors, hallmarked by low AR activity, neuroendocrine-like gene expression, minimal copy number alterations, and few mutations.

## Discussion

Here, we present a pioneering report on transcriptomics- and epigenetics-based subtyping of prostate cancer. As AR is the key-driving transcription factor in prostate cancer, tightly controlled by epigenetic regulation, integrating these datastreams has the potency to reveal distinct biological features for a tumor type classically-known as difficult to classify.

AR transcriptional regulation requires a permissive epigenetic environment, but the interdependent relationships thereof in clinical samples remained thus far elusive. We show here that unsupervised patient classification based on genome-wide profiles for AR, H3K27ac, and H3K27me3 (but not H3K4me3) groups patients in strong accordance to RNA-seq-based tumor profiling. Previously, we and others have revealed distinct AR chromatin binding profiles between healthy tissue and primary prostate cancer[17] as well as between primary prostate cancer and castration resistant prostate cancer (CRPC)[18], which successfully stratified patients on outcome. As no differences are observed in AR binding profiles between cases and controls in this study, we conclude that AR reprogramming is associated with tumor onset and disease progression, but no distinct AR profiles are found in primary disease that bear prognostic potential. In order to identify differences in the location of transcription factor binding sites and epigenetic marks between samples groups, genomic regions-of-interest need to be defined. In selecting high-confidence peaks that were reliably identified in multiple samples, we selected for each datatype its own cut-off in overlap between samples (based on Fig. 2f). As no clear consensus exists currently in the field on the minimal number of samples in which peaks should be identified (peaks found in at least 3 out of 8 samples[18]; union of peaks[13,19]; peaks found in at least 2 out of 21 samples[33]; peaks found in ~50% of samples[34]), future studies should be directed in

finding the optimal selection criteria while appreciating inter-tumor heterogeneity.

The *TMRPSS2-ERG* gene fusion is found in ~50% of all primary prostate cancer samples, and provided the most-potent patient stratification based on transcriptomic, AR cistromic, and epigenetic classification. This is in accordance with a recently reported smaller study using H3K27ac ChIP-seq data from 19 samples[13]. We confirm these findings in a far-larger dataset, and illustrate that AR and H3K27me3 show comparable profiles, indicating that *TMPRSS2-ERG* fusion may have far larger consequences beyond enhancer usage, dictating AR cistromic profiles along with genomic selectivity of polycomb-mediated gene suppression.

Integrative clustering, combining RNA-seq data with ChIP-seq for AR and the three histone modifications (H3K4me3, H3K27ac, H3K27me3) revealed three distinct prostate cancer subtypes. While *TMPRSS2-ERG* status still was a major discriminating feature in the integrative clustering analyses, we identified a third thus far unknown prostate cancer subtype that contained both high and low ERG expression samples. This subtype is hallmarked by benign features, with low AR chromatin binding and limited activity, although the receptor was readily expressed. Furthermore, this subtype is copy number-neutral with low mutation burden. Based on PAM50 classification, this subtype appeared selectively depleted from luminal B tumors, which were previously reported to be associated with poor outcome[14].

In clinical practice, PSA level is used as a biomarker for AR action. However, even though the tumors of cluster 3 had levels of both AR and PSA that were comparable to the other subtypes, AR activity score and AR chromatin binding were low in these samples. These data suggest not only that conventional biomarkers would support an incorrect conclusion in this setting, but also indicate that the tumors in cluster 3 may be potentially dictated by other tumor-promoting cascades than AR, such as FGF, WNT, and NGF pathways. Small molecule inhibitors targeting FGF have been developed, future clinical trials on FGF inhibitors may benefit from pre-selective patients from this cluster.

Interestingly, luminal B tumors—depleted in cluster 3—have been previously reported to have a better response to ADT as compared to non-luminal B tumors. Consequently, ADT may be least-effective in this specific patient subpopulation. In contrast, tumors in cluster 2 showed high AR activity score, indicating that these patients may derive benefit from ADT. However, RNA-seq data from a large cohort of patients receiving adjuvant ADT is required to test this hypothesis.

With this, we reveal that by integrating transcriptomic, epigenetic, and AR cistromic datastreams, previously unknown prostate cancer subtypes can be found with distinct biological and clinical features. Furthermore, these data will provide an invaluable resource for the community, presenting the largest prostate cancer dataset to date containing matched genomic, transcriptomic, and epigenetic datastreams. Future analyses may further expand our understanding of the interrelationships between these factors and how these could be affected by genomic and epigenetic queues.

## Methods

**Cohort.** Primary prostate cancer specimens from radical prostatectomy resections were collected at the Portuguese Oncology Institute, Porto, Portugal. From this cohort, 49 samples (cases) were selected from patients who developed a relapse within ~5 years after diagnosis and matched on age, Gleason score, PSA level, and T-stage with 50 samples (controls) from patients with non-relapsed disease within ~10 years after diagnosis. All samples included were histological diagnosed as clinically localized acinar adenocarcinoma of the prostate. Two samples had no follow-up data available. Fresh frozen material was trimmed to maximize tumor percentage (>70%) and cut in 30-micron sections for ChIP-seq or 10-micron

sections for RNA-seq. This study was approved by the institutional review board (Comissão de Ética para a Saúde, CES-IPOP-198/2012). All procedures performed in this study were in accordance with the ethical standards of the institutional and/or national research committee and with the 1964 Helsinki declaration and its later amendments or comparable ethical standards. Informed consent was obtained from the participants included in the study. A subset of the cohort was previously characterized for ETS transcription factor rearrangements[12].

**ChIP-seq.** Chromatin immunoprecipitations were performed as described previously with minor changes[35]. In brief, samples were crosslinked in solution A with 2 mM DSG (CovaChem) for 25 min at room temperature. After 25 min, 1% formaldehyde was added for 20 min and subsequently quenched with glycine. Samples were lysed as described[36] and sonicated for at least 10 cycles of 30 s on, 30 s off using a Diagenode Bioruptor Pico. For each ChIP, 5 μg of antibody was conjugated with 50 μl Protein A magnetic beads. Antibodies used were AR (sc-816, Santa Cruz), H3K27ac (39133, Active Motif), H3K4me3 (Ab8580, Abcam), and H3K27me3 (39155, Active Motif). Immunoprecipitated DNA was processed for library preparation (Part# 0801-0303, KAPA Biosystems kit). Libraries were sequenced using an Illumina Hiseq2500 genome analyzer (65 bp, single end), and aligned to hg19 using BWA (v0.5.10). Reads with a mapping quality >20 were selected. Peak calling over input control (mixed inputs) was performed using DFilter (v1.5)[37] and MACS[38] for AR and H3K27ac ChIP-seq samples. MACS 1.4 was run with *p*-value cutoff of 10e−7 and DFilter with bs = 50, ks = 30, refine, nonzero. For H3K4me3, MACS2 and DFilter were used with broad-peak settings: (1) –broad and –broad-cutoff = 0.2 for MACS2 and (2) bs = 100 and ks = 60 for DFilter. The peaks called by both peak callers were used for analysis. H3K27me3 ChIP-seq peaks were called by genome segmentation using ChromHMM (v1.12) choosing the state with high H3K27me3 signal[39]. Normalized strand coefficient (NSC) and relative strand correlation (RSC) were calculated using phantompeak-tools (Supplementary Data 1 and Supplementary Figure 3)[40].

Samples that passed the following quality parameters were included in the final analysis; tumor cell percentage ≥70%, ChIP-qPCR enrichment, more than 100 peaks called and NSC, RSC values higher than input samples.

Genome browser snapshots were generated using Easeq (v1.03)[41], motif analysis was performed using the Galaxy Cistrome SeqPos motif tool with default settings[42] and genomic region enrichment analysis was performed with CEAS[43]. Consensus peaklists were generated with the DiffBind R package (v2.4.6)[33]. BEDTools (v2.25) was used to calculated read counts in peaks[44]. The raw counts were normalized for library size followed by TMM normalization using EdgeR[45].

**RNA-seq.** RNA was extracted using the AllPrep DNA/RNA universal kit (Qiagen) according to the manufacturer's instruction with Qiagen's QIAcube robot. RNA quantity and quality were assessed with the 2100 Bioanalyzer using a Nano chip (Agilent, Santa Clara, CA) and samples with a RIN > 8 were considered for library preparation. Strand-specific libraries were generated with the TruSeq Stranded mRNA sample preparation kit (Illumina, Part # 15031047 Rev. E) and sequenced on a HiSeq2500. Sequencing data was aligned to hg38 using TopHat (v2.1.0 using bowtie 1.1.0) and number of reads per gene were measured with HTSeq count (v0.5.3). EdgeR (v3.18.1) -Limma (3.34) workflow was used for gene expression analysis[45,46]. The ComBat normalization method within the R package SVA (3.24.4) was used to correct the batch effects observed by day of RNA isolation[47]. Genes with >1 count per million in at least 10 samples were included.

For detecting ERG fusion transcripts, we used STAR-fusion (v0.5.4) with suggested parameters except for chimSegmentMin = 5, chimJunctionOverhangMin = 5, alignSJDBoverhangMin = 5, alignMatesGapMax = 200000, and alignIntronMax = 200000[48]. Coverage in ERG exones were measured using BEDtools (v2.25), followed by taking ratio between average expression of five exones close to 5′ end (chr21:39,870,287–39,870,428, chr21:39,947,586–39,947,671, chr21:39,956,768–39,956,869, chr21:40,033,582–40,033,704, and chr21:40,032,446–40,032,591 in hg19) and the exone at 3′ end (chr21:39,751,950–39,755,845 in hg19). A ratio of 0.3 was chosen to determine low or high 5′–3′ transcript ratio.

Tumor samples with mRNA expression data available were classified into luminal A, luminal B, or basal-like subtype using the PAM50 classifier. PAM50 clustering was performed as previously described[14]. The expression of one of the PAM50 genes (*MIA*) was not included in the Porto analysis. To calculate the CD49f signature score, normalized log2 expression value for each gene (91 genes) was multiplied by their weights and summed up[49]. AR activity score was calculated by the composite expression of 20 genes[24]. In addition, expression of 70 neuroendocrine signature genes were obtained from castration resistant neuroendocrine and prostate adenocarcinoma samples, previously published[26]. The expression of 14 of the 70 neuroendocrine signature genes were not included in the analysis for the Porto Cohort (*KIAA0408, SOGA3, KCNB2, KCND2, LRRC16B, NRSN1, PCSK1, RGS7, SEZ6, ST8SIA3, SVOP, PRR5-ARHGAP8, UPK2, MYCN*) and four genes for the TCGA cohort (*SOGA3, BRINP1, MAP10, PIEZO1*) because the genes are expressed only in <~10% of the samples. Expression fold changes between neuroendocrine tumors and adenocarcinoma samples were calculated. Concordance in expression differences (fold change sign) were measured using Pearson correlation.

**Cluster analysis**. Cluster analysis was performed using ConsensusClusterPlus (v1.40.0)[23]. The samples were clustered using the top 3000 most variable genes or the top 1000 most variable regions across the samples as determined by median absolute deviation. Consensus clustering was run by hierarchical clustering algorithm for 100 iterations with a resampling rate of 80%. For integrative clustering, we applied MIV-NMF[50] using the top 3000 most varying regions and genes as determined by median absolute deviation from the five data types. For each data type, we constructed a 3000 by 97 matrix in which rows are features and columns are samples. The order of samples is maintained across the data types, and 0 values were used for missing samples. The MIV-NMF takes the five matrices at the same time, factorizes them to obtain five sample factor matrices and finds a consensus sample factor matrix, which in turn also affects the individual sample factor matrix. Given a consensus sample factor matrix, the factor with the highest weight is identified for each sample, followed by grouping the samples with the same factor identified. Following regularization parameters are used for MIV-NMF: alpha = 0.001 and beta = 0. MIV-NMF was applied 20 times with random initialization and a consensus matrix capturing the stability of the clustering was constructed to identify the optimal number of clusters. Final clusters were defined by hierarchical clustering on the consensus matrix that captures stable associations across the 20 experiments. To further assess the stability of the three chosen clusters, the same clustering was repeated for 100 times with 80% subsampling, followed by constructing the same consensus matrix with the stability measured. To assess the contribution of each data type, Pearson correlation was measured for each individual sample factor matrix and consensus sample factor matrix. The statistical relation between consensus clusters and variables was tested using linear regression and Pearson's Chi-square test for continuous (age, PSA level, and AR activity scores) and categorical parameters (Gleason score, T-stage, ERG status, PAM50 subtype, and Group (Case/Control)), respectively.

**Copy number analysis**. The CopywriteR R package (v2.6.1) was used to extract DNA copy number information from off-target reads in AR ChIP-seq data[27]. The code is available on GitHub: https://github.com/PeeperLab/CopywriteRCustomBed. CopywriteR was run with 400 kb bins and a custom peak file using MACS2[38]. Furthermore, peaks were extended with 1 kb on either side to minimize the number of on-target reads. Downstream analysis of median normalized log2 values was performed using Circular Binary Segmentation (CBS) and the R package CGHcall[51]. To calculate copy number alteration burden per sample, number of bins with copy number gains or losses were determined and divided by the total number of bins.

**TCGA data**. Gene expression, protein expression, and somatic mutation data from the TCGA prostate cancer cohort were downloaded from https://xenabrowser.net/. For RNA-seq, genes with >1 count per million in at least 43 samples were included. TCGAbiolinks[52] was used to retrieve copy number data, followed by converting the data into genome-wide copy number profiles with 400 kb bin resolution. Tumor cellularity information and initial PSA values for 333 primary tumors were reported previously[8].

**Gene expression-based enrichment analysis**. Differentially expressed genes between the clusters were identified by comparing one cluster with the other two using limma. Given gene expression data with cluster label, gene set enrichment analysis was performed with ggsea R package (v1.0) (https://doi.org/10.5281/zenodo.438018) using ggsea_s2n (signal to noise ratio) and ggsea_weighted_ks (weighted KS statistic) as score function and enrichment score function, respectively. For each cluster, expression of genes was compared between one cluster versus the other two. Curated gene set collection was obtained from MsigDB v6.1. For each cluster in TCGA, top gene sets with significant enrichment with FDR < 0.2 are included for radar plot visualization. Differential regulatory networks were identified using HotNet2 (https://github.com/raphael-group/hotnet2). For initial heat, FDR score was obtained from differential gene expression analysis with limma. For reference protein–protein interaction (PPI) network, we used the three PPI networks of the original study[30]. After identifying the enriched PPI components in each of the three PPI networks, the components were merged with those sharing the common nodes.

**Immunohistochemistry**. Immunohistochemistry was performed using Bond™ Polymer Refine Detection kit (Leica Biosystems, Germany), in Leica Bond III platform. Antigen retrieval was performed for 10 min, with Epitope Retrieval Solution 1 for chromogranin and Epitope Retrieval Solution 2 for synaptophysin. Primary monoclonal antibodies for chromogranin (Clone DAK-A3, 1:1000 dilution, Dako, Denmark) and synaptophysin (Clone DAK-SYNAP, 1:100 dilution, Dako, Denmark) were used. Hematoxylin was used for nuclear counterstaining. Appropriate positive controls were used for each antibody and negative controls consisted on the omission of primary antibodies. Because all cases analyzed contained non-neoplastic prostatic epithelium, this served as (internal) control as appropriate.

Chromogranin and synaptophysin immunoexpression were assessed using a routine optical microscope by an experienced pathologist blinded to molecular data. Each marker was categorized as <1% positive neoplastic cells, 1–10% positive neoplastic cells and thereafter at 10% increments. Any cytoplasmic staining regardless of intensity was considered positive.

## Data availability

All ChIP-seq and RNA-seq data generated in this study are deposited in the Gene Expression Omnibus (GEO) database under the accession numbers GSE120738 and GSE120741, respectively. Public ChIP-seq datasets used in this study are available from GEO or the European Genome-phenome Archive under the following accession code: GSE70079 (AR ChIP-seq) and EGAS00001002496 (H3K27ac ChIP-seq).

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

## Acknowledgements

We would like to acknowledge the NKI-AVL Core Facility Molecular Pathology & Biobanking (CFMPB) for lab support, the NKI Genomics Core Facility for Illumina sequencing and bioinformatics support, and the NKI Research High Performance Computing (RHPC) facility for computational infrastructure. This work was supported by funding from Movember (NKI01), KWF Dutch Cancer Society (10084 ALPE), KWF Dutch Cancer Society/Alpe d'HuZes Bas Mulder Award (NKI 2014-6711), and a VIDI grant (016.156.401) from The Netherlands Organisation for Scientific Research (NWO).

## Author contributions

W.Z., A.M.B., S.S., and E.N. conceived the study. C.J. and R.H. performed sample collection and processing pathology assessment. J.L., C.J. and R.H. provided clinical information. S.S., K.S. and E.V.E. conducted the experiments. E.N., Y.K. and S.S. computationally analyzed the data. O.K. and D.S.P. provided support for copy number alterations analysis using ChIP-seq data. S.L.C. and F.Y.-C.F. were involved in PAM50 subtyping. W.Z., A.M.B., and L.F.A.W. provided scientific input and supervised the project. S.S., W.Z. and Y.K. wrote the manuscript with input from all authors.

## Additional information

**Competing interests:** The authors declare no competing interests.

