## [Peer Review File · Nature Communications]

Reviewers' comments:

Reviewer #1 (Remarks to the Author):

Prostate cancer is one of the most common cancer types. There have been several attempts to stratify prostate cancer patients based on genetic, genomic or morphological features. For example, prostate cancer patients can be stratified with TMPRSS2-ERG fusion or mutation in FOXA1, SPOP1 and IDH1. However, stratification based on epigenetic data has been less explored. This study aims to integrate epigenetics and gene expression data to stratify 100 prostate cancer patients, either with biochemical recurrence or not. The authors did not find any distinct difference in gene expression, AR binding or active histone modification marks between the two groups. However, when integrated, these datasets classify the patients in three subclasses, with a novel subclass hallmarked by low AR activity, neuroendocrine like gene expression and low copy number alterations.

Overall, this is a well-designed study and the dataset is a rich resource for epigenetic and prostate cancer research. A few comments need to be addressed.

Major points:

1. Quality control of the ChIP-seq data was glossed over. A table summarizes the ChIP-seq data (e.g. sequencing depth, mappability, number of peaks, fold change over input) would be very helpful.
2. A few previous studies have profiled AR and or H3K27ac (e.g. Pomerantz et al. Nature Genetics 2015; Kron et al. Nature Genetics 2017) in primary prostate cancer tumours, with smaller sample size. A comparison to these datasets would be useful for quality control.
3. The identification of cluster 3 with low AR activity and some NE signature is interesting. Is there any difference of histology (e.g. presence of small cell) of the cluster3 tumours compared to cluster 1/2?
4. Since cluster 3 has low AR-activity, it may be less responsive to ADT and may not have rising PSA. Overall survival or metastatic free survival should be more relevant for this cluster than the biochemical relapse free survival.

Minor

1. Page4, "25 out of 56 H3K4me3 samples and 15 out of 77 H3K27me3 samples for further analysis". What is the criterion of choosing the cutoff for each dataset?
2. Fig 5f is suppose to show gene sets with FDR < 0.2. However, in the corresponding supplementary table 2, as shown by column "fdr_TCGA", only the first two gene sets have fdrs less than 0.2. The p-values do not matched in the Fig 5f and supplementary table 2. For example, the p-value of "TOMLINS_PROSTATE_CANCER_UP" set seems to be near 1e-04 in Fig 5f TCGA, but is 0.041 for "p_TCGA" in supplementary table 2. In addition, supplementary table 2 only shows gene sets significantly enriched (FDR < 0.2) in TCGA cohort. They should also show gene sets significantly enriched (FDR < 0.2) in Porto cohort.

Reviewer #2 (Remarks to the Author):

In the manuscript by Stelloo et al, the authors performed RNA-seq with ChIP-seq for AR and histone modification marks (H3K27ac, H3K4me3, H3K27me3) in 100 primary prostate carcinomas. Integrative molecular subtyping of the five data streams revealed three major subtypes of which two were clearly TMPRSS2-ERG dictated. Their analysis discovered a third novel subtype, with low chromatin binding and activity of AR, but with high activity of FGF and WNT signaling. This group was positive for neuroendocrine-hallmark genes, copy number-neutral with low mutational burden, and significantly depleted for genes characteristic of poor-outcome associated luminal B-subtype.

This is an interesting study with solid data which will be highly valuable for the research community. Although this is a well written manuscript, I do have the following concerns:

1. The authors should discuss the clinical significance and potential impact for clinical practice.
2. Any evidence that any of the neuroendocrine markers is expressed at the protein level?

Reviewer #1 (Remarks to the Author):

Prostate cancer is one of the most common cancer types. There have been several attempts to stratify prostate cancer patients based on genetic, genomic or morphological features. For example, prostate cancer patients can be stratified with TMPRSS2-ERG fusion or mutation in FOXA1, SPOP1 and IDH1. However, stratification based on epigenetic data has been less explored. This study aims to integrate epigenetics and gene expression data to stratify 100 prostate cancer patients, either with biochemical recurrence or not. The authors did not find any distinct difference in gene expression, AR binding or active histone modification marks between the two groups. However, when integrated, these datasets classify the patients in three subclasses, with a novel subclass hallmarked by low AR activity, neuroendocrine like gene expression and low copy number alterations.

Overall, this is a well-designed study and the dataset is a rich resource for epigenetic and prostate cancer research. A few comments need to be addressed.

We thank the reviewer for the comments and suggestions, and we are delighted to hear the reviewer appreciates our work as a rich resource for the community.

Major points:

1. Quality control of the ChIP-seq data was glossed over. A table summarizes the ChIP-seq data (e.g. sequencing depth, mappability, number of peaks, fold change over input) would be very helpful.

We agree that additional information about quality control of our ChIP-seq experiments would surely contribute to the paper, and we thank the reviewer for highlighting this. The existing boxplots illustrate total read count, percentage aligned reads, phantom peak results and number of called peaks in all samples sequenced (Supplementary Figure S3a-d (previously S13a-d)). We have now added a table (Supplementary Table S1) with information on the following parameters for all samples included in the analysis.

- Total number of reads
- Percentage aligned reads
- number of reads mapped as single read
- Normalized strand cross correlation (NSC, phantompeaktools)
- Relative strand cross correlation (RSC, phantompeaktools)
- Number of peaks
- fraction of reads in peaks (FRiP).

Included in the results section (line 91): *'Information on ChIP-seq quality metrics is summarized in Supplementary Table S1 and Supplementary Figure S3.'*

With this, we believe we have included all relevant quality control parameters, and we hope the reviewer agrees.

2. A few previous studies have profiled AR and or H3K27ac (e.g. Pomerantz et al. Nature Genetics 2015; Kron et al. Nature Genetics 2017) in primary prostate cancer tumours, with smaller sample size. A comparison to these datasets would be useful for quality control.

Thank you for the suggestion. We have addressed this question with three different analysis;

1) Quality metrics

To compare quality metrics between our data and both publicly available datasets, we used NSC and RSC values. These analyses illustrated our data to perform comparably to the currently available small cohort studies. These metrics, a measure of enrichment independent of peak calling algorithms and/or thresholds, have been used in large consortia such as ENCODE and ROADMAP. We have added the results in the first paragraph of the results sections and represented the results in Supplementary Figure S3e-f.

Added to the results section (line 92-96): *'To validate the quality metrics of ChIP-seq, we utilized publicly available AR ChIP-seq (n=13) and H3K27ac ChIP-seq (n=19) on primary prostate cancers to compare normalized strand cross correlation (NSC) and relative strand cross correlation (RSC) values. Overall similar RSC and NSC values were observed among the datasets, with all NSC values from the ChIP-seq samples higher than input samples (Supplementary Figure S3e-f).'*

In addition, we calculated the fraction of reads in peaks (FRiP) for the H3K27ac ChIP-seq data as also input samples were provided. Overall the FRiP scores were higher in the previously published dataset as compared to our data, plausibly due to read depth, which was on average 8 times higher as compared to this study. Since the paper from Pomerantz et al. (Nature Genetics, 2015) did not provide input samples for peak calling, FRiP scores could not be generated using the identical settings/QC criteria as we used in our study in which input was used, as FRiP scores are affected by the peak calling approach that was used (Nakato and Shirahige, Brief Bioinform 2017).

We now added to the results section (line 96-100): *'The fraction of reads in peaks (FRiP) score for H3K27ac in our study were lower as compared to those reported by Kron et al. Nature Genetics 2017 (Supplementary Table S2), which is possibly related to our lower sequencing depth (Nakato and Shirahige, Brief Bioinform 2017). As no input files were available for Pomerantz et al. (Nature Genetics 2015), peak calling analyses could not be performed uniformly among both datasets, which affects FRiP score (Nakato and Shirahige, Brief Bioinform 2017).'*

2) ChIP-seq signal at the consensus sites that were defined in this study

As an additional quality control analysis for our data, we now provide a new analysis in which signal for publicly available data on AR (Pomerantz et al.) and H3K27ac (Kron et al.) was analysed at the consensus binding sites we identified. At these consensus sites, robust and strong signal was found for all samples from the publicly available datasets (Supplementary Figure S5). We added this in the results section (line 123-127): *'We used publicly available AR ChIP-seq and H3K27ac ChIP-seq data to check for enrichment of these factors at the consensus sites defined in this study. At these 8162 AR consensus sites and 31085 H3K27ac consensus sites, robust and strong signal was found for all samples from the publicly available datasets (Supplementary Figure S5).'*

3) Samples separate on ERG status

In addition, we analyzed the ChIP-seq signal of the previously published H3K27ac ChIP-seq data, interrogating the ERG-status-differential H3K27ac sites defined in this study. In addition, we performed the reciprocal analyses, in which we analyzed the ChIP-seq signal from our Porto samples at the sites clustering patients on ERG status as defined in the Kron et al. paper. For both cohorts a clear separation on ERG status was observed. Added to the results section (line 172-175): *'We also found separation of ERG fusion positive and negative samples using previously published H3K27ac ChIP-seq data looking at the sites defined in this study (Supplementary Figure S10a). The sites defined in the previous study, also classified the Porto samples on ERG expression (Supplementary Figure S10b).'*

Unfortunately, ERG status of the public AR ChIP-seq data has never been determined (personal communication prof. Matthew Freedman).

3. The identification of cluster 3 with low AR activity and some NE signature is interesting. Is there any difference of histology (e.g presence of small cell) of the cluster3 tumours compared to cluster 1/2?

These are all primary PCa cases, treated with radical prostatectomy and correspond to conventional acinar adenocarcinoma of the prostate. Pathological analyses revealed no foci of small cell carcinoma in any of the cases. Small cell carcinoma is relatively rare in the prostate and usually corresponds to a form of progression after androgen-deprivation therapy and/or radiotherapy. Consequently, it is very unusual to find neuroendocrine prostate cancer as a primary diagnosis. Even in those cases, and due to the aggressiveness of the tumor, it is usually locally advanced and/or metastatic and, thus, not eligible for radical prostatectomy. No difference on histology was observed among the samples. We have added the following text to the methods section (line 359-360): *'All samples included were histological diagnosed as clinically localized acinar adenocarcinoma of the prostate.'*

While all tumors we analyzed were not neuroendocrine cancers, we did identify cluster 3 as positive for some, but not all, neuroendocrine-like features: 1) Even though AR was expressed in cluster 3, it did not appear functional 2) we found enrichment of a neuroendocrine gene signature.

As suggested by this reviewer, as well as reviewer 3 (comment 2), we performed immunohistochemical staining for two neuroendocrine markers; chromogranin and synaptophysin. However, these two classical NEPC markers were not selectively enriched in cluster 3, further strengthening our conclusion that while our results suggest some neuroendocrine features as relatively enriched in cluster 3, these cancers are still adenocarcinomas nonetheless.

4. Since cluster 3 has low AR-activity, it may be less responsive to ADT and may not have rising PSA. Overall survival or metastatic free survival should be more relevant for this cluster than the biochemical relapse free survival.

We indeed hypothesized that the clusters may be predictive for response to androgen deprivation therapy (ADT); ADT may be least effective for patients classified in cluster 3. Unfortunately, to our knowledge, no other sizeable cohort of ADT-treated patients based on fresh frozen/RNA-seq data is available, which prevents us to address this question directly. Nonetheless, to try and resolve this issue, we instead applied our clustering based on the 285 genes to an FFPE Affymetrix patient cohort published by Zhao et al, JAMA Oncol, 2017. Unfortunately, we were not able to find comparable clustering. It is commonly accepted in the field that translating classifiers between different platforms (RNA-seq versus microarray) or sample work-up (fresh-frozen versus FFPE) is challenging and affects classifier performance. We now updated the discussion section (line 340-344): *'Consequently, androgen deprivation therapy may be least-effective in this specific patient subpopulation. In contrast, tumors in cluster 2 showed high AR activity score, indicating that these patients may derive benefit from androgen deprivation therapy. However, RNA-seq data from a large cohort of patients receiving adjuvant ADT is required to test this hypothesis'*

We fully agree that overall survival or metastatic free survival is more relevant as end-point, rather than biochemical relapse free survival. However, both in our cohort as well as the TCGA cohort, number of events on metastatic relapse or overall survival are limited. For the TCGA, we now included the recurrence free survival information (Table 3), but not overall survival due to limited events (n=10). For the Porto cohort, we have now included overall and metastatic survival in table 2. In total, 21 patients developed a clinical recurrence, 23 patients died of which 8 died of prostate cancer. Between the three

clusters, no statistically significant differences could be observed on overall survival and/or metastatic free survival, which could very well be related to the low number of events and thus limited statistical power.

Added in the results section (line 220-223): *'In the Porto cohort, out of the 97 patients, 21 patients developed a clinical recurrence, 23 patients died of which 8 died of prostate cancer. Between the three subgroups, no statistically significant differences could be observed on metastatic free survival and overall survival, which may be related to the low number of events.'*

Minor

1. Page4, "25 out of 56 H3K4me3 samples and 15 out of 77 H3K27me3 samples for further analysis".
What is the criterion of choosing the cutoff for each dataset?

These decisions are made on the desire to work with high confidence peaks as mentioned in the results; *'To focus on the high-confidence peaks that were reproducibly identified in a large number of tumors'*. In the field, there are thus far no strict guidelines on cut-offs or consistency of peaks identified between samples; previously we used the peaks present in 3 out of 8 tumors (Stelloo et al, 2015), Kron et al, (Nat gen, 2017) and Pomerantz et al, (Nat gen, 2015) for example studied the union of all peaks ever found in any sample, while Ross-innes et al, (Nature, 2012) limited their analyses on the peaks found in at least 2 out of the 21 samples. In our recently ChIP-seq based studies in male breast cancer (Severson et al., Nature Comm, 2018), we used as cut-off all peaks identified in at least 50% of the samples analyzed, which is more stringent.

As shown in figure 2f, we observed a steep decrease in the number of overlapping peaks as the number of samples increases for AR and H3K27me3 ChIP-seq samples. Therefore, we chose peaks present in at least ~25% of the samples, while less heterogeneity was observed for the two active histone marks H3K27ac and H3K4me3. Hence we chose peaks present in at least ~45% of the samples. These cut-offs were based on the shape of the curve in Figure 2f, aimed to focus the analysis on highly-reproducible peaks while maintaining sufficient variation between datasets to identify novel biological features.

Added in the discussion section (line 309-316): *'In order to identify difference in the location of transcription factor binding sites and epigenetic marks between samples groups, genomic regions-of-interest need to be defined. In selecting high-confidence peaks that were reliably identified in multiple samples, we selected for each datatype its own cut-off in overlap between samples (based on Figure 2f). As no clear consensus exists currently in the field on the minimal number of samples in which peaks should be identified (Stelloo et al, (EMBO mol med, 2015): peaks found in at least 3 out of 8 samples; Kron et al, (Nat gen, 2017) and Pomerantz et al, (Nat gen, 2015): union of peaks; Ross-innes et al, (Nature, 2012): peaks found in at least 2 out of 21 samples; Severson et al., (Nature Comm, 2018): peaks found in ~50% of samples), future studies should be directed in finding the optimal selection criteria while appreciating inter-tumor heterogeneity.'*

2. Fig 5f is suppose to show gene sets with FDR < 0.2. However, in the corresponding supplementary table 2, as shown by column "fdr_TCGA", only the first two gene sets have fdrs less than 0.2. The p-values do not matched in the Fig 5f and supplementary table 2. For example, the p-value of "TOMLINS_PROSTATE_CANCER_UP" set seems to be near 1e-04 in Fig 5f TCGA, but is 0.041 for "p_TCGA" in supplementary table 2. In addition, supplementary table 2 only shows gene sets significantly

enriched (FDR < 0.2) in TCGA cohort. They should also show gene sets significantly enriched (FDR < 0.2) in Porto cohort.

We apologize for the mistake in Supplementary Table S2 (in revised manuscript S4). We have now corrected the table, listing (as originally intended) the significantly enriched genesets with FDR <0.2 in either the TCGA cohort and Porto cohort. Due to the larger sample size in TCGA, many of the gene sets are significant in the TCGA cohort and not in the Porto cohort. However, direction of normalized enrichment score (NES) shows concordance in biological regulation between the TCGA and Porto cohorts.

Added to the results section (line 275-277): *'Even though most of the gene sets reached significance only in the TCGA cohort, most likely due to smaller sample size of the Porto cohort, the normalized enrichment score (NES) between the Porto and TCGA cohort correlated well.'*

Please note that we noticed an error in the original version of Figure 2c, which has now been corrected. The re-analysis did not affect any of the conclusions drawn throughout the manuscript.

In addition we noticed an textual error: we used the top 1000 most variable regions for consensus clustering of H3K27ac data and not top 3000 as previously written.

Reviewer #2 (Remarks to the Author):

In the manuscript by Stelloo et al, the authors performed RNA-seq with ChIP-seq for AR and histone modification marks (H3K27ac, H3K4me3, H3K27me3) in 100 primary prostate carcinomas. Integrative molecular subtyping of the five data streams revealed three major subtypes of which two were clearly TMPRSS2-ERG dictated. Their analysis discovered a third novel subtype, with low chromatin binding and activity of AR, but with high activity of FGF and WNT signaling. This group was positive for neuroendocrine-hallmark genes, copy number-neutral with low mutational burden, and significantly depleted for genes characteristic of poor-outcome associated luminal B-subtype.

This is an interesting study with solid data which will be highly valuable for the research community. Although this is a well written manuscript, I do have the following concerns:

We are happy to hear the reviewer appreciates our work, and finds the data highly valuable for the research community. We attempted to address the issues raised in full, as further elaborated below.

1. The authors should discuss the clinical significance and potential impact for clinical practice.

We thank the reviewer for this comment. As recommended, we now updated a paragraph in the discussion section (line 332-344), further discussing the clinical significance of our findings;

'In clinical practice, PSA level is used as a biomarker for AR action. However, even though the tumors of cluster 3 had levels of both AR and PSA that were comparable to the other subtypes, AR activity score and AR chromatin binding were low in these samples. These data suggest not only that conventional biomarkers would support an incorrect conclusion in this setting, but also indicate that the tumors in cluster 3 may be potentially dictated by other tumor-promoting cascades than AR, such as FGF, WNT and NGF pathways. *As small molecule inhibitors targeting FGF have been developed, future clinical trials on FGF inhibitors may benefit from pre-selecting patients from this cluster.*

Interestingly, luminal B tumors -depleted in cluster 3- have been previously reported to have better response to androgen deprivation therapy as compared to non-luminal B tumors. Consequently, androgen deprivation therapy may be least-effective in this specific patient subpopulation. In contrast, tumors in cluster 2 showed high AR activity score, indicating that these patients may derive benefit from androgen deprivation therapy. *However, RNA-seq data from a large cohort of patients receiving adjuvant ADT is required to test this hypothesis'*

2. Any evidence that any of the neuroendocrine markers is expressed at the protein level?

Firstly, on the basis of histology the samples all represent acinar adenocarcinoma of the prostate, with no foci of small cell carcinoma in any of the cases. We mention this now in the methods section (line 359). Based on these data and the AR-positivity of all samples studied, and in line with the known low frequency of primary neuroendocrine prostate cancers, we concluded that the samples we studied were not neuroendocrine tumors. Nonetheless, some features were found in cluster 3 that could be considered as 'neuroendocrine-like properties', including an absence of AR functionality (even though the receptor was expressed) and positive enrichment of a neuroendocrine gene signature. As suggested by the reviewer, we performed immunohistochemistry (IHC) for the detection of two classical neuroendocrine markers chromogranin and synaptophysin on all Porto samples used for integrative clustering (n=97). The

markers chosen are typical neuroendocrine markers used in clinical practice. Description of IHC staining and assessment is now included in the methods (line 485-497).

Most cases have less than 1% of cells expressing chromogranin and/or synaptophysin. In addition, the number of cases that stain positive (>1%) for chromogranin (n=10) are fewer than those that stain for synaptophysin (n=22). This would be expected as chromogranin is more specific and synaptophysin is more sensitive, as markers of neuroendocrine differentiation. Even though synaptophysin IHC results are consistent with the RNA-seq based data for this marker (Supplementary Figure S15c), we don't observe association of the mRNA expression and protein expression for these neuroendocrine markers with cluster 3. The results are shown in Supplementary Figure S15.

We clarify that by no means we want to claim that subtype 3 is neuroendocrine prostate cancer, but rather represents tumors that possess a number of features that overlap with neuroendocrine prostate cancer, including the absence of AR signaling. For each subtype, we compared the relative expression (i.e. fold change in the subtype relative to the others) of NEPC signature genes. With this, we found a NEPC-associated gene expression profile relatively enriched in subtype 3 as compared to subtype 1 and 2, but not in an absolute sense. As independent assessment of potential NEPC status on IHC, we thus selected the two markers that are standard-of-care in clinical practice, which are not represented in the NEPC signature. With these results, we ruled out cluster 3 as genuine NEPC.

To enhance clarity on this issue, we now reformulated the description of these findings, with a stronger focus on a lack of AR-signaling in cluster 3, and a limited representation of some NEPC-genes of mRNA levels. Furthermore, as FGF signaling was found enriched in cluster 3, and small molecule inhibitors for FGFs have been developed and tested in clinical trials, the discussion section now further elaborates on the potential clinical applicability of our findings in clinical trial design.

We state now the following in the results section (line 242-253): *'To detect neuroendocrine differentiation on the protein level, we performed immunohistochemistry (IHC) for the detection of two neuroendocrine markers typically used in clinical practice, chromogranin and synaptophysin. As both genes (CHGA and SYP) are not represented in the neuroendocrine gene expression signature, these findings would be an independent validation of NEPC-status in these tumors. However, most cases have less than 1% of cells expressing chromogranin and/or synaptophysin (Supplementary Figure S15a). Only 10 samples stained positive (>1%) for chromogranin and 22 samples for synaptophysin. Importantly, both CHGA and SYP were not enriched in cluster 3 at either the mRNA level and protein level (Supplementary Figure S15b-c). Thus, while cluster 3 was enriched for tumors that possess a number of neuroendocrine-like features, including lack of AR functionality (even though the receptor is expressed) and enrichment for a neuroendocrine gene signature, classical pathological neuroendocrine markers were negative in these tumors.'*

Please note that we noticed an error in the original version of Figure 2c, which has now been corrected. The re-analysis did not affect any of the conclusions drawn throughout the manuscript.

In addition we noticed a textual error: we used the top 1000 most variable regions for consensus clustering of H3K27ac data and not top 3000 as previously written.

REVIEWERS' COMMENTS:

Reviewer #1 (Remarks to the Author):

All of my previous comments have been addressed satisfactorily. I do not have further comments.

Reviewer #2 (Remarks to the Author):

I am happy with the revised manuscript. It is acceptable for publication